# TravelReasoner: Reasoning-Augmented Travel Survey Simulations with Large Reasoning Models

## Abstract

Travel survey plays a central role in a wide range of applications, such as urban planning and traffic management. Large language models (LLMs) have recently demonstrated huge potential in simulating human behaviours. However, previous works in travel survey simulation research have primarily focused on tuning LLMs to directly fit travel survey data, overlooking the underlying reasoning process behind human decision-making. The emergence of large reasoning models (LRMs) has achieved tremendous success in solving complex tasks, offering unique opportunities to simulate a realistic travel survey through LLM reasoning. In this paper, we introduce *TravelReasoner*, a novel framework that enhances travel survey simulations by integrating the reasoning capabilities of LRMs. We construct *Chain-of-Trips* from publicly available trip-chain records in the National Household Travel Survey (NHTS). This dataset captures the step-by-step reasoning process behind real-world travel decisions. To improve the accuracy and rationality of LRMs' in-domain reasoning, we propose a post-training pipeline via curriculum learning. Experiments demonstrate that TravelReasoner substantially outperforms strong baselines, location consistency improved by 6.8%, and time consistency improved by 4.1%, while producing interpretable intermediate reasoning traces that enable transparent and explainable simulations. Our findings highlight the promise of LRMs for complex decision modeling and open new opportunities for applying NLP to urban systems. Data and code are available at https://anonymous.4open.science/r/TravelReasoner-4037

## 1 Introduction

Understanding human travel behavior is essential for designing efficient transportation systems, guiding urban planning, and supporting evidence-based policy evaluation (Handy, 1996). Travel surveys remain a primary tool in this domain (Stopher & Greaves, 2007). However, traditional travel surveys face significant challenges (Westat, 2018), including low response rates, high collection costs, and insufficient contextual data. These limitations hinder their scalability and reliability, particularly in dynamic urban environments. As a result, there is a pressing need for effective alternatives that can efficiently generate realistic travel data. In addition, several studies have demonstrated the feasibility of using large language models(LLMs) to simulate human behaviorPark et al. (2023); Gao et al. (2024); Piao et al. (2025), offering a promising direction for travel simulation.

Recent advances have explored the use of LLMs to generate synthetic travel surveys (Li et al., 2024; Bhandari et al., 2024; Zhang & Xu, 2025). Current LLM-based simulation approaches typically rely on training with large-scale text data, enabling models to replicate human travel patterns. Despite their promise, these methods still fall short in several areas, particularly in capturing the deep reasoning and complex behaviors underlying human travel decisions (Bhandari et al., 2024). While LLMs can generate plausible activity sequences, they often fail to account for the intricate decision-making processes behind travel choices, leading to simulations that lack behavioral realism and interpretability. The emergence of large reasoning models (LRMs) has enhanced model performance on complex reasoning tasks (Xu et al., 2025), and integrating their reasoning capabilities with simulations has enabled the capture of more realistic travel trajectories.

In this paper, we introduce TravelReasoner, a novel framework for enhancing travel simulation using LRMs. The core of our approach lies in improving travel survey simulations by integrating the advanced reasoning capabilities of LRMs. Unlike prior research, which treats trips as discrete sequences, we reframe travel chains as chains of thought. To support this, we construct the Chain-of-Trips dataset, derived from the National Household Travel Survey (NHTS) data. This dataset captures the intricate causal, temporal, and motivational structures behind travel decisions by simulating a first-person perspective. We address several critical questions—whether, why, when, where, and how—step by step, reflecting the reasoning process behind each decision. Additionally, we propose a two-stage post-training pipeline, combined with supervised fine-tuning, to optimize behavioral authenticity and first-person coherence in the generated travel narratives.

We validate our approach through extensive experiments on multiple city-level simulation tasks, benchmarking it against traditional methods and an existing LLM baseline. Our experimental results show that the proposed TravelReasoner significantly outperforms the strongest baseline, location consistency improved by 6.8% and time consistency improved by 4.1%. Moreover, our model demonstrates strong cross-domain generalization, as it performs well across datasets from different cities, highlighting its broad applicability.

The key contributions of this work are as follows:

- We are the first to apply LRMs to travel behavior modeling and survey simulation. And we introduce the Chain-of-Trips dataset based on real-world NHTS data, which captures multi-level reasoning patterns in travel decision-making.
- We propose a two-stage post-training pipeline that combines supervised fine-tuning to enhance the model's reasoning capabilities and the fidelity of generated behaviors.
- Our extensive experiments demonstrate the advantages of our approach in terms of reasoning plausibility, behavioral consistency, and cross-domain generalization.

## 2 RELATED WORKS

### 2.1 TRAVEL SURVEY SIMULATION

Simulating travel surveys has long been pursued as a cost-effective alternative to traditional data collection, which often suffers from high costs, privacy concerns, and low response rates(Greaves & Stopher, 2000; Mattson, 2012; Administration, 2017). Early approaches employed Monte Carlo sampling based on decision tree clustering of households to model trip attributes(Greaves & Stopher, 2000; Stopher & Pointer, 2004), later extended using neural networks to enhance transferability across regions(Mohammadian et al., 2010), albeit with limited success in capturing temporal or modal details. Agent-based models (ABMs) simulate travel behavior by modeling individuals with synthetic needs and preferences(Kim et al., 2019; 2020), effectively generating location-based social network data, though often limited in capturing diverse or rare activity chains.

Recent research has turned to large language models (LLMs) for survey simulation, leveraging their capacity to encode common-sense and contextual knowledge from large corpora(Brown et al., 2020; Petroni et al., 2019; JIAWEI et al., 2024; Wang et al., 2023). LLMs have been shown to predict next destinations and generate human-like activity sequences. (Bhandari et al., 2024) proposes a LLM-based framework that prompts models to generate synthetic travel diaries. Their evaluation—at pattern, trip, and activity-chain levels—demonstrates that fine-tuned LLMs can outperform both base models and agent-based simulations, producing data that closely resembles actual survey distributions even in cities unseen during training.

### 2.2 REASONING WITH LARGE LANGUAGE MODELS

Reasoning with large language models (LLMs) has emerged as a central focus in recent NLP research. While early LLMs demonstrated strong capabilities in language understanding and pattern completion, they often lacked explicit multi-step reasoning abilities required for tasks such as commonsense inference, planning, and decision modeling(Ouyang et al., 2022). To bridge this gap, recent work has explored various prompting strategies—such as Chain-of-Thought prompting(Wei et al., 2022; Liu et al., 2025)—that elicit step-by-step reasoning traces from LLMs. Further advances

introduced Interaction-of-Thought(Zhao et al., 2025), a method that simulates reasoning as multi-agent interaction, improving coherence and factual accuracy in complex tasks like recommendation and planning.

Several efforts fine-tune LLMs on domain-specific reasoning corpora to improve transferability and robustness(Ouyang et al., 2022; Hu et al., 2022). This includes training models on structured reasoning tasks (e.g., math word problems, game states) and aligning outputs through reinforcement learning with feedback (Ouyang et al., 2022; Shao et al., 2024b). Such techniques have shown success in enhancing both reasoning quality and interpretability, crucial for high-stakes domains like healthcare, law, and urban planning. However, their application to travel survey simulation—where decisions involve personal constraints, preferences, and sequential dependencies—remains under-explored.

Building on prior work, we present a reasoning-augmented LLM framework for simulating travel behavior. In contrast to existing LLM-based simulators that primarily capture surface-level correlations, our approach explicitly models reasoning traces through supervised fine-tuning and reinforcement-based feedback. This enables more behaviorally realistic and interpretable simulations, bridging the gap between statistical accuracy and transparent decision modeling.

## 3 TRAVELREASONER

In this section, we present **TravelReasoner**, a reasoning-augmented framework for travel survey simulation. We first provide an overview of the simulation process, then describe the construction of the *Chain-of-Trips* dataset from real NHTS data, and finally detail our two-stage training paradigm designed to enhance reasoning and improve generalization in travel survey simulation.

### 3.1 OVERVIEW

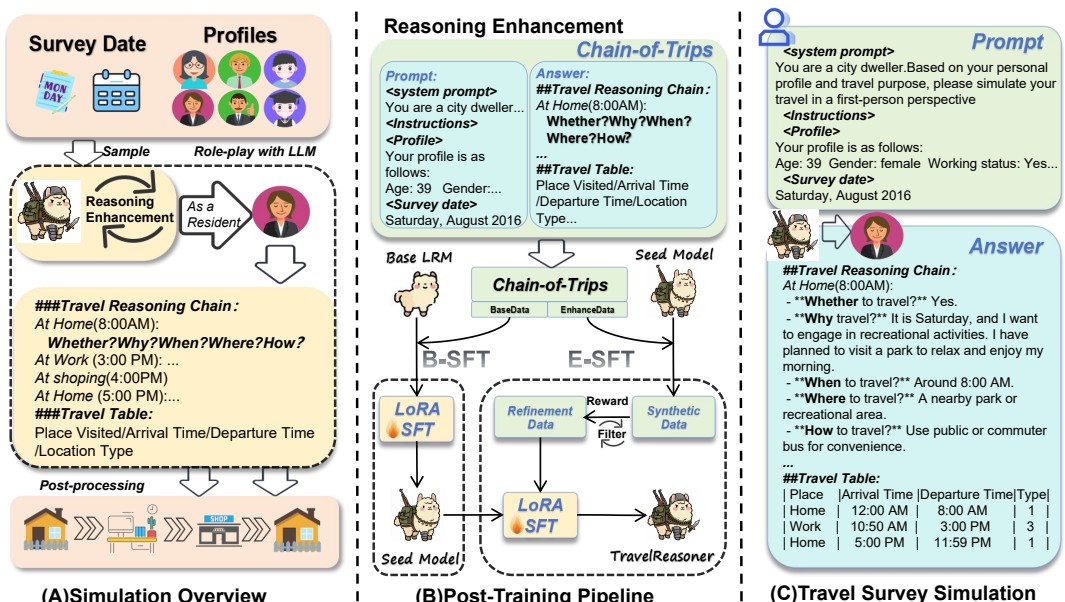

Figure 1: Overview of the TravelReasoner framework. It consists of: (A) travel data generation based on profiles and survey dates, (B) post-training pipeline via two-stage supervised fine-tuning, and (C) first-person simulation of travel decisions.

The overall design of TravelReasoner is illustrated in Figure 1. In (A), given a sampled survey date and a user profile (e.g., age, gender, employment status) from the NHTS dataset, the model is prompted to assume the role of a city resident and simulate daily travel behavior from a first-person perspective. The simulation yields two complementary outputs: (1) a *Travel Reasoning Chain*, which captures sequential decision-making at each time step (e.g., whether to travel, why, where,

when, and how), and (2) a structured *Travel Table*, recording trip attributes such as location type, arrival time, and departure time. Post-processing the Travel Table reconstructs a complete activity chain, providing a realistic mobility trajectory. This design enables the model to generate not only plausible trip sequences but also interpretable reasoning aligned with human decision processes, producing synthetic survey data suitable for urban planning, transportation modeling, and behavioral analysis.

Figure 1(B) illustrates the two-stage pipeline designed to enhance reasoning capabilities. In the first stage, we perform supervised fine-tuning on a portion of the Chain-of-Trips dataset, enabling the model to learn structured reasoning patterns based on real-world behavior. In the second stage, we use another portion of the dataset to generate answers for the fine-tuned model in the first stage, and then select high-quality question-answer pairs for the second stage of fine-tuning. These pairs are then used for additional fine-tuning, allowing the model to learn through self-reinforcement and encouraging it to generate logically consistent trip chains.

Figure 1(C) presents an example after training. Given a user profile and a survey date, the model simulates detailed travel behavior from a first-person perspective. At each time point, it explicitly reasons through core behavioral questions—whether to travel, why, where, when, and how—producing natural language justifications alongside structured trip records. This demonstrates the model's ability to generate interpretable, goal-directed, and contextually grounded travel behavior.

## 3.2 CHAIN-OF-TRIPS CONSTRUCTION

To support reasoning-augmented travel modeling, we construct *Chain-of-Trips*, a structured dataset derived from the NHTS. Each instance represents a single day of travel decisions from a first-person perspective, conditioned on contextual factors such as demographics, activity purposes, and temporal constraints.

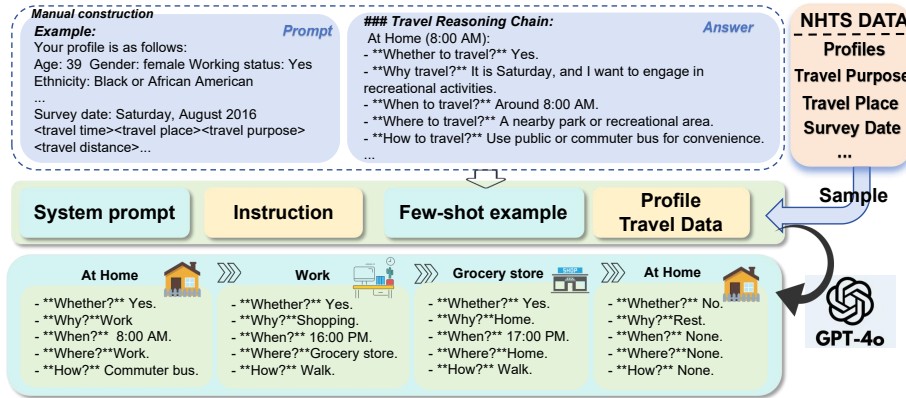

Figure 2: Construction of the *Chain-of-Trips* dataset from NHTS data. User profiles and travel records are extracted to build structured prompts with few-shot examples. GPT-4o then generates step-by-step reasoning chains under realistic decision contexts.

As shown in Figure 2, we first sample user profiles and daily travel logs from NHTS, including attributes such as age, gender, employment status, ethnicity, and survey date, along with trip sequences. We organize this information into structured prompts with four components: (1) a system prompt specifying the simulation objective, (2) task instructions defining the reasoning scope, (3) a manually constructed few-shot example demonstrating the reasoning structure, and (4) the target user profile with contextual details. Few-shot examples, curated from real NHTS patterns, are critical for guiding interpretable, multi-step decision trajectories.

Given this setup, a language model (GPT-4o) generates a *Travel Reasoning Chain*, reflecting step-by-step decisions (whether to travel, why, where, when, and how), and a parallel *Travel Table*, recording structured trip attributes such as location type, arrival time, and departure time. Each dataset instance is represented as a triplet $(Q, R, A)$: the prompt $Q$ contains the system message, task instruction, and user profile; the reasoning component $R$ is the generated *Travel Reasoning Chain*; and the answer $A$ is the corresponding *Travel Table*. This triplet provides the supervision

signal for training. By jointly modeling travel sequences and the reasoning behind them, *Chain-of-Trips* enables the model to learn realistic mobility patterns together with human-aligned decision logic. This dual-format representation—natural language reasoning plus structured outputs—offers a rich training signal for grounding LLMs in the travel behavior domain. Detailed prompts can be found in Appendix A.3.

## 3.3 POST-TRAINING PIPELINE

To enhance the model's ability to simulate human-like travel reasoning, we employ a two-stage training pipeline combined with LoRA ((Hu et al., 2022)), a parameter-efficient adaptive method. In the first stage, we fine-tune the model using Chain-of-Trips data to enhance its structured reasoning capabilities. In the second stage, we applied self-learning to improve the model's inference stability and further enhance its inference quality.

### 3.3.1 SUPERVISED FINE-TUNING

We fine-tune the model on the *Chain-of-Trips* dataset using LoRA, which freezes the pretrained weights and introduces a pair of trainable low-rank matrices into each target layer. Formally, instead of updating the full weight matrix $W_0 \in \mathbb{R}^{d \times k}$, LoRA parameterizes the weight update as:

$$\Delta W = AB, \quad A \in \mathbb{R}^{d \times r}, B \in \mathbb{R}^{r \times k}, r \ll \min(d, k), \tag{1}$$

where $r$ is the low-rank dimension. The effective weight becomes $W = W_0 + \Delta W$, while only $A$ and $B$ are trainable. This design enables efficient fine-tuning with orders-of-magnitude fewer trainable parameters compared to full-parameter updates.

Each training instance is represented as a triplet $(Q, R, A)$, where $Q$ is the prompt, $R$ the reasoning chain, and $A$ the structured answer table. We concatenate $(Q, R, A)$ as the target output and optimize the standard auto-regressive language modeling objective:

$$\mathcal{L}_{\text{SFT}} = -\sum_{t=1}^{T} \log P_\theta(y_t \mid y_{<t}, Q), \tag{2}$$

where $y_t$ denotes the $t$-th token in the combined target $(R, A)$, and $\theta$ are the LoRA-augmented model parameters.

### 3.3.2 TWO-STAGE TRAINING PIPELINE

We propose a two-stage training pipeline that incorporates the principles of curriculum learning, consisting of a Base Supervised Fine-Tuning (B-SFT) stage and an Enhanced SFT (E-SFT) stage.

In the B-SFT stage, we fine-tune the LRM using a portion of the Chain-of-Trips dataset, enabling it to grasp the fundamental paradigms and preliminary reasoning capabilities of the domain task, thereby constructing a baseline model.

In the E-SFT stage, we design an iterative self-optimization process to enhance the model's performance further. First, we use the LRM obtained in the B-SFT stage to generate a large number of candidate samples. Then, through manual screening and high-quality data curation (human-in-the-loop curation), our screening metric is shown in Equation 3. We screen the top 20% of question-answer pairs in the answers to construct a small, high-quality "golden" dataset. Finally, we use this refined dataset for a second round of fine-tuning, resulting in the final reasoning model, TravelReasoner. This stage significantly improves the model's reasoning capabilities by integrating the model's generation capabilities with human prior knowledge.

$$\text{Reward} = A \cdot \exp\left(-\frac{\text{AverLoc}}{\alpha}\right) + B \cdot \exp\left(-\frac{-\text{TimeCons}}{\beta}\right) + C \cdot \exp\left(-\frac{\text{EditDist}}{\gamma}\right) \tag{3}$$

Where AverLoc represents the Mean Absolute Error (MAE) between the generated chain and the actual chain, TimeCons represents the Root Mean Square Error (RMSE) between the generated stay time and the actual time, EditDist represents the edit distance between the generated chain and the actual chain. $A$, $B$, and $C$ are the indicator weights, and $\alpha$, $\beta$, and $\gamma$ are the scaling factors. Here, we use $A = 0.2$, $B = 0.4$, $C = 0.4$, $\alpha = 2$, $\beta = 60$, and $\gamma = 2$.

# 4 EXPERIMENTAL SETUP

## 4.1 DATASET

We base our study on the 2017 NHTS Trip Chaining Dataset[*], a large-scale survey conducted by the U.S. Federal Highway Administration. The dataset provides comprehensive, real-world records of individual travel behavior, including trip-level information such as departure and arrival times, trip purposes, and visited locations, as well as detailed sociodemographic profiles of participants (e.g., age, gender, race, education level, employment status, and household income). These rich attributes make the dataset particularly suitable for modeling reasoning-aware travel behavior.

For our purposes, the NHTS data serves two roles. First, it provides the foundation for constructing the *Chain-of-Trips* dataset, where individual travel trajectories are reformulated into structured prompts and reasoning chains. Second, it supports evaluation, allowing us to benchmark the plausibility and coherence of simulated outputs against realistic human travel behavior. This dual role enables both robust model training and meaningful empirical validation.

## 4.2 IMPLEMENTATION DETAILS

We utilize DeepSeek-R1-Distill-Llama-8B as our experimental model, setting the temperature to 0.6 and top-p to 1. In line with the configurations in SigSpatial (Bhandari et al., 2024), we restrict travel locations to 20 categories. Our experiments leverage real-world NHTS data and carefully curated question-answer pairs, conducted across four cities(San Francisco, San Diego, Austin, Atlanta). During the B-SFT phase, we fine-tune the model using Low-Rank Adaptation with the Adam optimizer, a learning rate of 1e-4, and 2000 training samples. In the E-SFT phase, we used the inference outputs of the model trained in phase 1 on an additional 1,000 training examples and selected 200 high-quality inference data points for this phase of training. More details can be found in Appendix A.1.

## 4.3 BASELINES

We used the following methods as baselines. These methods leverage the LLM's ability to process and reason about complex, semantically rich data to generate and predict mobility behaviors. These methods are more flexible and adaptable, and can handle a variety of tasks by combining human-like reasoning and contextual understanding, such as V-LRM(vanilla LRM), LRM-CoT(Wei et al., 2022), Bhandari24(Bhandari et al., 2024), CoPB(Shao et al., 2024a), and LLMob(JIAWEI et al., 2024).

- **V-LRM**: Represents a vanilla LRM, an untrained version of the model that has not yet undergone any specialized training or fine-tuning.
- **LRM-CoT**: Utilizes large language models to simulate mobility, enhancing the generation of mobility intentions by incrementally breaking down reasoning processes.
- **Bhandari24**: A model focused on spatially-augmented generation, which incorporates geographic factors and personal preferences to simulate mobility behavior.
- **CoPB**: A workflow that integrates the Theory of Planned Behavior into mobility behavior generation, incorporating attitudes, subjective norms, and perceived behavioral control to improve the accuracy of mobility predictions.
- **LLMob**: An LLM agent framework that accounts for individual activity patterns and motivations, employing a self-consistency approach to align LLMs with real-world activity data, and a retrieval-augmented strategy for interpretable activity generation.

## 4.4 EVALUATION METRICS

To comprehensively evaluate the quality of simulated travel behavior, we adopt three complementary metrics. These metrics assess accuracy at the trip level, temporal consistency, and sequence-level similarity. Together, they provide a holistic evaluation of both individual trajectories and aggregated mobility patterns. See Table 4.4 for a detailed description of the metrics.

---

[*]https://nhts.ornl.gov/

| Metric | Formula | Description |
|--------|---------|-------------|
| **AverLoc** | $\frac{1}{N}\sum_{i=1}^{N}|\hat{\ell}_i - \ell_i|$ | The mean absolute error (MAE) between the generated chain and the actual chain length. |
| **TimeCons** | $\sqrt{\frac{1}{N}\sum_{i=1}^{N}(\hat{t}_i - t_i)^2}$ | Root mean square error (RMSE) of the dwell time between the generated and actual chains. |
| **EditDis** | $\frac{1}{N}\sum_{i=1}^{N}\mathrm{Lev}(\hat{s}_i, s_i)$ | Edit distance between the generated chain and the actual chain. |

Specifically, $\hat{\ell}_i$ and $\ell_i$ denote the predicted and ground-truth location categories, respectively. $\hat{t}_i$ and $t_i$ represent predicted and actual stay durations. Finally, $\hat{s}_i$ and $s_i$ are the predicted and ground-truth location sequences, and $\mathrm{Lev}(\cdot)$ denotes the Levenshtein distance.

## 5 RESULTS AND ANALYSIS

### 5.1 ZERO-SHOT AND FEW-SHOT

Table 1 presents the results of our preliminary experimental analysis. We perform comparative experiments using both the baseline model and the semantic reasoning model in zero-shot and few-shot scenarios.

|  | Model | AverLoc | TimeCons | EditDis |
|--|-------|---------|----------|---------|
| Zero-shot | Base | 21.04 | 349.03 | 23.36 |
|  | Reasoning | 4.01 | 145.33 | 6.29 |
| Few-shot | Base | 2.07 | 126.61 | 3.92 |
|  | Reasoning | 2.33 | 114.99 | 4.31 |

Table 1: Experiments with the base model and reasoning model in zero-shot and few-shot scenarios. For few-shot scenarios, we manually construct chain-of-thought samples. The base model used is Llama-3.1-8B, and the reasoning model is DeepSeek-R1-Distill-Llama-8B.

The preliminary results indicate that the reasoning model outperforms the baseline in zero-shot settings, providing strong justification for its use in subsequent experiments. In the few-shot scenario, we carefully designed three Chain-of-Thought examples to facilitate model learning through imitation. The introduction of CoT examples substantially enhances the imitation capabilities of both models, validating the construction of the Chain-of-Trip dataset for training generalization models.

### 5.2 MAIN RESULTS

In this section, we present the key experimental results of TravelReasoner and compare its performance with well-known baselines, including V-LRM, LRM-CoT, CoPB, LLMob, and Bhandari24, using the AverLoc, TimeCons, ODSim, and EdiDis metrics introduced in Section 4.4.

The results, presented in Tables 2 and 7, demonstrate that TravelReasoner consistently achieved either the best or second-best performance across all evaluation metrics. For instance, on the San Francisco dataset, TravelReasoner recorded an AverLoc of 1.85, outperforming the strong baseline Bhandari24 (1.91). Furthermore, it achieved the best results in terms of temporal consistency (TimeCons = 91.88) and sequence edit distance (EditDis = 2.84). Similarly, across datasets from three additional cities, TravelReasoner outperformed all other methods, highlighting its robustness in diverse urban contexts. These findings underscore that TravelReasoner not only generates accurate trip sequences but also preserves high temporal rationality and behavioral consistency, thereby validating the efficacy of our reasoning-enhanced approach in travel simulation. On average, TravelReasoner improves location consistency by 6.8% and time consistency by 4.1% compared to the strongest baseline.

|  | San Francisco | | | San Diego | | |
|---|---|---|---|---|---|---|
|  | **AverLoc** | **TimeCons** | **EditDis** | **AverLoc** | **ODSim** | **EditDis** |
| V-LRM | 4.01 | 145.33 | 6.29 | 4.08 | 156.39 | 6.22 |
| LRM-CoT | 3.69 | 147.60 | 5.95 | 3.94 | 151.80 | 6.06 |
| CoPB | 6.72 | 198.04 | 9.12 | 5.81 | 175.12 | 8.40 |
| LLMob | 2.74 | 131.22 | 5.09 | 2.80 | 128.02 | 5.02 |
| Bhandari24 | 1.91 | 96.40 | 3.17 | 1.94 | 97.62 | 3.06 |
| TravelReasoner | **1.85** | **91.88** | **2.84** | **1.90** | **89.90** | **2.85** |

Table 2: Performance comparison of TravelReasoner with the baseline model on the San Francisco and San Diego datasets. Bold indicates the best result, and underlined indicates the second-best result. V-LRM represents a vanilla LRM, an untrained LRM.

In addition to the overall performance, we also analyze the results of TravelReasoner for different demographic groups in the San Francisco dataset, including age, gender, and income categories. As shown in Table 3 and Table 6, the model demonstrates consistent performance across these groupings. Specifically, TravelReasoner achieved AverLoc of 1.77/1.93 for males and females, and 1.69/1.93 for younger (under 40) and older (40 and over) groups. 1.73/1.93 in the low-income and high-income groups, respectively, and 1.77/1.94. Whether in spatial accuracy (AverLoc), temporal consistency (TimeCons), or sequence edit distance (EditDis), TravelReasoner outperforms or approaches the best baseline Bhandari24. These findings suggest that TravelReasoner is not only effective at generating accurate travel sequences at a city-wide level but also exhibits a strong ability to adapt to various demographic profiles, further validating the robustness and versatility of our reasoning-enhanced approach in diverse urban contexts.

|  | Male | | | Female | | |
|---|---|---|---|---|---|---|
|  | **AverLoc** | **TimeCons** | **EditDis** | **AverLoc** | **TimeCons** | **EditDis** |
| V-LRM | 4.12 | 149.1 | 6.28 | 3.91 | 141.72 | 6.31 |
| LRM-CoT | 3.65 | 147.96 | 5.83 | 3.74 | 147.25 | 6.08 |
| CoPB | 6.91 | 196.13 | 9.22 | 6.54 | 199.90 | 9.02 |
| LLMob | 2.93 | 127.08 | 5.17 | 2.56 | 135.39 | 5.02 |
| Bhandari24 | 1.94 | 96.21 | 3.21 | **1.87** | 96.58 | 3.14 |
| TravelReasoner | **1.77** | **88.43** | **2.72** | 1.93 | **95.19** | 2.96 |

Table 3: Performance comparison of TravelReasoner and baseline models on different groups(gender) on the San Francisco dataset. Bold indicates the best result, and underlined indicates the second-best result. v-LRM represents a vanilla LRM, an untrained LRM.

## 5.3 CROSS-CITY GENERALIZATION

To validate the model's cross-domain generalization, we used data from four cities (San Francisco, Austin, San Diego, and Atlanta) for training and tested it on Dallas-Fort Worth and Los Angeles (see Table 4).

Experimental results show that TravelReasoner maintains its strong performance in novel cities, maintaining its lead over other baselines in AverLoc and EditDis. For example, on the Dallas–Fort Worth dataset, TravelReasoner achieved an AverLoc score of 1.87 and an EditDis score of 2.67, both outperforming Bhandari24 (1.95/2.93). It also achieved the best results on the Los Angeles dataset (AverLoc = 1.85, EditDis = 2.79), with significant improvements in temporal consistency. This shows that the model can not only learn reasonable travel patterns in the training city, but also be transferred to unseen urban scenes, showing good cross-domain generalization ability. This ability is crucial for real-world travel simulation because practical applications often require the transfer of models between different cities without the need for a large amount of local annotated data.

| | Dallas-Fort Worth | | | Los Angeles | | |
|---|---|---|---|---|---|---|
| | AverLoc | TimeCons | EditDis | AverLoc | ODSim | EditDis |
| V-LRM | 3.93 | 143.65 | 6.13 | 4.03 | 142.21 | 6.21 |
| LRM-CoT | 3.97 | 137.28 | 6.12 | 4.16 | 144.19 | 6.32 |
| CoPB | 5.49 | 191.60 | 7.80 | 2.54 | 143.18 | 4.88 |
| LLMob | 2.97 | 130.91 | 5.27 | 6.13 | 189.60 | 8.38 |
| SigSpatial | 1.95 | 89.70 | 2.93 | 1.91 | 102.08 | 3.12 |
| TravelReasoner | **1.87** | **82.99** | **2.67** | **1.85** | **94.50** | **2.79** |

Table 4: Our method generalizes to other cities. We train it using travel data from four cities (San Francisco, Austin, San Diego, Atlanta) and evaluate it using data from Dallas-Fort Worth and Los Angeles.

## 5.4 ABLATION STUDIES

To further assess the contribution of each module in our approach, we conducted ablation experiments using datasets from San Francisco and San Diego (see Table 5).

Compared to V-LRM, the inclusion of B-SFT resulted in significant improvements across all evaluation metrics, highlighting the crucial role of supervised fine-tuning in learning fundamental reasoning patterns. The introduction of E-SFT, based on a self-learning paradigm, further enhances model performance, demonstrating that the enhanced fine-tuning stage improves reasoning consistency and behavioral rationality through the incorporation of high-quality, human-curated samples. Overall, the two-stage training pipeline is synergistic, with both stages being indispensable. The fully integrated TravelReasoner outperforms the reduced version in terms of both accuracy and consistency.

| | San Francisco | | | San Diego | | |
|---|---|---|---|---|---|---|
| | AverLoc | TimeCons | EditDis | AverLoc | ODSim | EditDis |
| V-LRM | 4.01 | 145.33 | 6.30 | 4.08 | 156.39 | 6.22 |
| TR(w/o E-SFT) | 1.89 | 100.12 | 3.27 | **1.89** | 97.24 | 3.21 |
| TravelReasoner | **1.85** | **91.88** | **2.84** | 1.90 | **89.90** | **2.85** |

Table 5: This table shows the results of ablation experiments in San Francisco and San Diego. V-LRM represents a vanilla LRM, an untrained LRM.

## 6 CONCLUSION

In this work, we introduce TravelReasoner, a novel framework designed to enhance travel survey simulations through LRMs. By leveraging the reasoning capabilities of LRMs, we are able to simulate human travel behavior in a more interpretable and behaviorally plausible manner. The core of our approach is the Chain-of-Trips dataset, which enables the model to learn structured decision-making patterns from real-world travel data. Our post-training pipeline optimizes the model's ability to generate realistic, first-person travel simulations. Experimental results demonstrate that TravelReasoner outperforms baseline models in both accuracy and behavioral coherence, producing travel simulations that closely mirror human mobility patterns. Specifically, TravelReasoner improves location consistency by 6.8% and time consistency by 4.1% compared to the strongest baseline. Moreover, the reasoning traces generated by the model provide valuable insights into the underlying cognitive processes driving travel decisions. In future work, we aim to explore the integration of additional contextual data, such as environmental factors or real-time urban events, to refine the realism of our simulations further.

## ETHICS STATEMENT

This research uses the publicly available NHTS dataset, which is de-identified and does not compromise personal privacy. The synthetic travel data generated by TravelReasoner is intended for research and policy analysis, not for individual profiling. We adhere to research integrity standards and do not involve direct human experimentation or sensitive information.

## REPRODUCIBILITY STATEMENT

We describe the data construction, training process, and evaluation methods in detail in the main text and appendix, and provide a link(`https://anonymous.4open.science/r/TravelReasoner-4037`) to an anonymized code repository to facilitate replication of experimental results and validation of model performance.

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

# A APPENDIX

## A.1 IMPLEMENTATION DETAILS

**Dataset Quality**  For the NHTS data, we first performed an initial filtration by removing trips with fewer than 3 or more than 10 locations for each city, and ensuring that no data contained three consecutive locations. During the creation of the chain-of-trips dataset, we employ a system prompt, instruction, few-shot examples, and task prompts (which can be found in Appendix A.3), and use the advanced closed-source model GPT-4 to construct the travel inference chain data.

**Detailed Experimental Parameter Setting**  During training and inference, we use the existing integrated Llama-factory for fine-tuning and Vllm for efficient inference, respectively.

Below are examples of Llama-factory fine-tuning parameters and Vllm inference parameters.

Llama-factory Fine-tuning Parameters Example

```
CUDA_VISIBLE_DEVICES=xxx llamafactory-cli train \
  --stage sft \
  --do_train \
  --model_name_or_path
     ./model/lora/v11/DeepSeek-R1-Distill-Llama-8B-trained \
  --dataset train_travel_reasoning_data_enhance \
  --dataset_dir ./data \
  --template deepseekr1 \
  --finetuning_type lora \
  --lora_target
     q_proj,v_proj,k_proj,o_proj,up_proj,down_proj,gate_proj \
  --lora_rank 64 --lora_alpha 128 --lora_dropout 0.05 \
  --output_dir ./saves/DeepSeek-R1-Distill-Llama-8B/lora/large/sft \
  --overwrite_output_dir \
  --cutoff_len 4096 \
  --preprocessing_num_workers 16 \
  --per_device_train_batch_size 4 \
  --per_device_eval_batch_size 2 \
  --gradient_accumulation_steps 4 \
  --lr_scheduler_type cosine \
  --logging_steps 10 \
  --warmup_ratio 0.03 \
  --save_strategy steps \
  --save_steps 200 \
  --eval_steps 100 \
  --do_eval \
  --eval_strategy steps \
  --load_best_model_at_end \
  --metric_for_best_model eval_loss \
  --greater_is_better false \
  --learning_rate 1e-4 \
  --num_train_epochs 5 \
  --val_size 0.1 \
  --plot_loss \
  --save_total_limit 3 \
  --bf16
```

Vllm Inference Parameters Example

```
    llm = LLM(model_name, tensor_parallel_size=2)
    sampling_params = SamplingParams(
        temperature=0.6,
        top_p=1,
```

```
        top_k=50,
        max_tokens=4096,
        repetition_penalty=1.0
    )
    response = llm.generate(prompt, sampling_params,use_tqdm=False)
```

**Compute Resources**   We train and infer LRM on two A100 GPUs with 80GB of RAM. Each experiment took from several minutes to several hours, depending on the number of training and test sets.

### A.2 SUPPLEMENTARY EXPERIMENTAL RESULTS

This section presents the supplementary experimental results of this paper.

**Group experiment(SF)** Table 6 presents a performance comparison between TravelReasoner and other baseline models on the San Francisco dataset, stratified by demographic groups, including gender (male vs. female), age (younger than 40 years vs. 40 years and older), and income (low income: household annual per capita income below $40,000; high income: household annual per capita income above $40,000).

| | Younger | | | Older | | |
|---|---|---|---|---|---|---|
| | **AverLoc** | **TimeCons** | **EditDis** | **AverLoc** | **TimeCons** | **EditDis** |
| V-LRM | 3.52 | 146.67 | 5.58 | 4.37 | 144.37 | 6.81 |
| LRM-CoT | 2.91 | 147.24 | 5.00 | 4.25 | 147.86 | 6.64 |
| CoPB | 7.01 | 199.24 | 9.20 | 6.52 | 197.22 | 9.06 |
| LLMob | 2.94 | 134.76 | 5.13 | 2.61 | 128.86 | 5.07 |
| Bhandari24 | 1.90 | 99.29 | 3.07 | **1.91** | 94.33 | 3.25 |
| TravelReasoner | **1.73** | **94.76** | **2.71** | 1.93 | **89.82** | **2.94** |

| | Low-income | | | High-income | | |
|---|---|---|---|---|---|---|
| | **AverLoc** | **TimeCons** | **EditDis** | **AverLoc** | **TimeCons** | **EditDis** |
| V-LRM | 3.36 | 139.56 | 5.53 | 4.74 | 153.48 | 7.15 |
| LRM-CoT | 4.06 | 145.18 | 6.2 | 3.41 | 153.27 | 5.80 |
| CoPB | 5.76 | 176.20 | 8.09 | 7.61 | 220.85 | 10.08 |
| LLMob | 2.83 | 128.54 | 5.10 | 2.66 | 135.03 | 5.09 |
| Bhandari24 | 1.80 | 91.65 | 2.98 | 2.02 | 102.57 | 3.40 |
| TravelReasoner | **1.77** | **90.14** | **2.68** | **1.94** | **95.19** | **3.02** |

Table 6: Performance comparison of TravelReasoner and baseline models on different groups on the San Francisco dataset. Bold indicates the best result, and underlined indicates the second-best result. v-LRM represents a vanilla LRM, an untrained LRM.

**Main Table Supplementary Experiments** Table 7 shows the performance comparison of Travel-Reasoner and other baseline models on the Atlanta and Austin datasets.

**Ablation study** Table 8 shows the results of ablation experiments in Austin and Atlanta.

| | Atlanta | | | Austin | | |
|---|---|---|---|---|---|---|
| | **AverLoc** | **TimeCons** | **EditDis** | **AverLoc** | **TimeCons** | **EditDis** |
| V-LRM | 3.92 | 133.76 | 5.98 | 4.98 | 154.24 | 7.12 |
| LRM-CoT | 3.38 | 138.93 | 5.51 | 3.76 | 141.58 | 5.90 |
| CoPB | 5.74 | 177.04 | 7.96 | 5.84 | 184.39 | 8.14 |
| LLMob | 2.87 | 136.33 | 5.09 | 2.81 | 137.24 | 5.06 |
| Bhandari24 | 1.81 | 93.79 | 2.76 | **1.75** | 92.11 | 2.85 |
| TravelReasoner | **1.77** | **88.38** | **2.65** | 1.77 | **90.09** | **2.71** |

Table 7: Performance comparison of TravelReasoner with the baseline model on the Atlanta and Austin datasets. Bold indicates the best result, and underlined indicates the second-best result. v-LRM represents a vanilla LRM, an untrained LRM.

| | Atlanta | | | Austin | | |
|---|---|---|---|---|---|---|
| | **AverLoc** | **TimeCons** | **EditDis** | **AverLoc** | **ODSim** | **EditDis** |
| V-LRM | 3.92 | 133.76 | 5.98 | 4.98 | 154.24 | 7.12 |
| TR(w/o E-SFT) | **1.75** | 100.02 | 3.02 | 1.83 | 96.12 | 3.10 |
| TravelReasoner | 1.77 | **88.38** | **2.65** | **1.77** | **90.09** | **2.71** |

Table 8: This table shows the results of ablation experiments in Austin and Atlanta. V-LRM represents a vanilla LRM, an untrained LRM.

## A.3 PROMPT

**Construction prompt** Here's a prompt example for constructing a chain of trips. This includes a system prompt, instructions, a few-shot example, and a target task.

```
system prompt

You are a city dweller. Based on your personal profile and travel
    purpose, please simulate your travel in a first-person
    perspective, construct a reasoning chain (whenever you are in a
    place, think about your travel plan to the next place,
    including whether to travel? Why travel? When to travel? Where
    to travel? How to travel (in terms of transportation)?), your
    travel should follow the Instructions content, and then
    generate your complete travel plan table (the table shows your
    stay time in each place, not the travel time).

The final output must follow the following table format:
| Visited Places | Arrival Time | Departure Time | Place Type |
|----------------|--------------|----------------|------------|
| [Place Name]   | [HH:MM AM/PM]| [HH:MM AM/PM]  | [Place Type]|
```

```
Instruction

Instructions:
1. If "home" is part of the travel activities on the specified
    date, please make sure to include it in the list.
2. The exact arrival and departure times recorded in the travel
    diary.
3. Enter the arrival time and departure time carefully, because a
    certain travel time needs to be maintained to ensure the
```

```
        rationality of the trip, and the arrival time of the current
        location is always later than the departure time of the
        previous location.
   4. Note that in the travel plan, the difference between the
        departure time of the previous location and the arrival time of
        the current location represents the travel time, and the
        difference between the arrival time and departure time of a
        location represents the stay time at that location.
   5. For [Location Type], please use only the numeric codes provided
        below:

   Location type code:
   1: Regular home activities (chores, sleep)
   2: Work from home (paid)
   3: Work
   4: Work-related meeting/trip
   5: Volunteer activities (not paid)
   6: Drop off/pick up someone
   7: Change type of transportation
   8: Attend school as a student
   9: Attend child care
   10: Attend adult care
   11: Buy goods (groceries, clothes, appliances, gas)
   12: Buy services (dry cleaners, banking, service a car, etc)
   13: Buy meals (go out for a meal, snack, carry-out)
   14: Other general errands (post office, library)
   15: Recreational activities (visit parks, movies, bars, etc)
   16: Exercise (go for a jog, walk, walk the dog, go to the gym, etc)
   17: Visit friends or relatives
   18: Health care visit (medical, dental, therapy)
   19: Religious or other community activities
   97: Something else
```

Few-Shot example

```
   Example:
   Your profile is as follows:
   Age: 39
   Gender: female
   Ethnicity: Black or African American
   Education: Some college or associates degree
   Working status: Yes
   Household annual income per capita: $42,500
   Place of residence: San Francisco
   Survey date: Saturday, August 2016

   You conducted the following travel activities on the survey date
        (travel time, travel origin-->travel destination, travel
        purpose, travel purpose summary, travel distance (miles),
        transportation method):

   1.8:00-9:00, Regular home activities (chores, sleep)-->Recreational
        activities (visit parks, movies, bars, movies, etc.),
        Social/Recreational, Home-based trip (social/recreational),
        15.056, Public or commuter bus
   2.10:00-10:50, Recreational activities (visit parks, movies, bars,
        movies, etc.) -->Work, Work, Not a home-based trip, 5.26,
        Public or commuter bus
   3.15:00-17:00, Work-->Regular home activities (chores, sleep),
        Home, Home-based trip (work), 20.409, Public or commuter bus
```

```
Please construct a travel reasoning chain from a first-person
    perspective based on your portrait and travel purpose (whenever
    you are at each location, think about traveling to the next
    location, including whether to travel? Why travel? When to
    travel? Where to travel? How to travel (in terms of
    transportation)? ), your travel should follow the Instructions
    content, and then generate your complete travel plan (this
    table shows your stay time in each place, not the travel time),
    as follows:
| Visited Places | Arrival Time | Departure Time | Place Type |
|----------------|--------------|----------------|------------|
| [Place Name]   | [HH:MM AM/PM]| [HH:MM AM/PM]  | [Place Type]|

Answer:
### Travel Reasoning Chain:

#### At Home (8:00 AM):
- **Whether to travel?** Yes.
- **Why travel?** It is Saturday, and I want to engage in
    recreational activities. I have planned to visit a park to
    relax and enjoy my morning.
- **When to travel?** Around 8:00 AM.
- **Where to travel?** A nearby park or recreational area.
- **How to travel?** Use public or commuter bus for convenience.

#### At Recreational Activity Location (10:00 AM):
- **Whether to travel?** Yes.
- **Why travel?** I have a work commitment later in the morning, so
    I need to leave the park and head to my workplace.
- **When to travel?** Around 10:00 AM to ensure I arrive at work on
    time.
- **Where to travel?** My workplace.
- **How to travel?** Use public or commuter bus.

#### At Work (3:00 PM):
- **Whether to travel?** Yes.
- **Why travel?** After finishing work, I want to head back home
    and relax for the rest of the day.
- **When to travel?** I will leave work at 3:00 PM to get home in
    the afternoon.
- **Where to travel?** Home.
- **How to travel?** Use public or commuter bus.

#### At Home (5:00 PM):
- **Whether to travel?** No.
- **Why travel?** After a full day of recreational and work
    activities, I need to prepare dinner and recharge for the
    upcoming workweek.
- **When to travel?** Not applicable.
- **Where to travel?** Not applicable.
- **How to travel?** Not applicable.

---

### Complete Travel Plan Table:

|Place Visited          |Arrival Time|Departure Time|Location Type|
|-----------------------|------------|--------------|-------------|
|Home                   | 12:00 AM   | 8:00 AM      | 1           |
|Recreational Activity  | 9:00 AM    | 10:00 AM     | 15          |
|Work                   | 10:50 AM   | 3:00 PM      | 3           |
|Home                   | 5:00 PM    | 11:59 PM     | 1           |
```

```
Target Task

Task:
Your profile is as follows:
Age: 77
Gender: male
Ethnicity: White
Education: Bachelor degree
Working status: No
Household annual income per capita: $112,500
Place of residence: San Francisco
Survey date: Tuesday, August 2016

You conducted the following travel activities on the survey date
    (travel time, travel origin-->travel destination, travel
    purpose, travel purpose summary, travel distance (miles),
    transportation method):

1.16:50-17:05, Regular home activities (chores, sleep)-->Buy goods
    (groceries, clothes, appliances, gas), Shopping/Errands,
    Home-based trip(shopping),2.712, Van

2.17:20-17:35, Buy goods (groceries, clothes, appliances,
    gas)-->Buy goods (groceries, clothes, appliances, gas),
    Shopping/Errands, Not a home-based trip, 1.494, Van

3.17:45-17:50, Buy goods (groceries, clothes, appliances,
    gas)-->Volunteer activities (not paid), Something else, Not a
    home-based trip, 0.853, Van

Please construct a travel reasoning chain from a first-person
    perspective based on your portrait and travel purpose (whenever
    you are at each location, think about traveling to the next
    location, including whether to travel? Why travel? When to
    travel? Where to travel? How to travel (in terms of
    transportation)? ), your travel should follow the Instructions
    content, and then generate your complete travel plan (this
    table shows your stay time in each place, not the travel time),
    as follows:
| Visited Places | Arrival Time | Departure Time | Place Type |
|----------------|--------------|----------------|------------|
| [Place Name]   | [HH:MM AM/PM]| [HH:MM AM/PM]  | [Place Type]|
```

**Chain-of-Trips example**    Here's a demonstration of the main question-answer pairs in the Chain-of-Trips.

```
Chain-of-Trips

You are a city dweller. Based on your personal profile and travel
    purpose, please simulate your travel in a first-person
    perspective, construct a reasoning chain (whenever you are in a
    place, think about your travel plan to the next place,
    including whether to travel? Why travel? When to travel? Where
    to travel? How to travel (in terms of transportation)?), your
    travel should follow the Instructions content, and then
    generate your complete travel plan table (the table shows your
    stay time in each place, not the travel time).

The final output must follow the following table format:
| Visited Places | Arrival Time | Departure Time | Place Type |
|----------------|--------------|----------------|------------|
```

```
| [Place Name]   | [HH:MM AM/PM]| [HH:MM AM/PM] | [Place Type]|

Instructions:
1. If "home" is part of the travel activities on the specified
   date, please make sure to include it in the list.
2. The exact arrival and departure times recorded in the travel
   diary.
3. Enter the arrival time and departure time carefully, because a
   certain travel time needs to be maintained to ensure the
   rationality of the trip, and the arrival time of the current
   location is always later than the departure time of the
   previous location.
4. Note that in the travel plan, the difference between the
   departure time of the previous location and the arrival time of
   the current location represents the travel time, and the
   difference between the arrival time and departure time of a
   location represents the stay time at that location.
5. For [Location Type], please use only the numeric codes provided
   below:

Location type code:
1: Regular home activities (chores, sleep)
2: Work from home (paid)
3: Work
4: Work-related meeting/trip
5: Volunteer activities (not paid)
6: Drop off/pick up someone
7: Change type of transportation
8: Attend school as a student
9: Attend child care
10: Attend adult care
11: Buy goods (groceries, clothes, appliances, gas)
12: Buy services (dry cleaners, banking, service a car, etc)
13: Buy meals (go out for a meal, snack, carry-out)
14: Other general errands (post office, library)
15: Recreational activities (visit parks, movies, bars, etc)
16: Exercise (go for a jog, walk, walk the dog, go to the gym, etc)
17: Visit friends or relatives
18: Health care visit (medical, dental, therapy)
19: Religious or other community activities
97: Something else

Task:
Your profile is as follows:
Age: 77
Gender: male
Ethnicity: White
Education: Bachelors degree
Working status: No
Household annual income per capita: $112,500
Place of residence: San Francisco
Survey date: Tuesday, August 2016

Please construct a travel reasoning chain from a first-person
   perspective based on your portrait and travel purpose (whenever
   you are at each location, think about traveling to the next
   location, including whether to travel? Why travel? When to
   travel? Where to travel? How to travel (in terms of
   transportation)? ), your travel should follow the Instructions
   content, and then generate your complete travel plan (this
   table shows your stay time in each place, not the travel time),
   as follows:
| Visited Places | Arrival Time | Departure Time | Place Type |
```

```
|----------------|--------------|----------------|------------|
| [Place Name]   | [HH:MM AM/PM]| [HH:MM AM/PM]  | [Place Type]|

Answer:
### Travel Reasoning Chain:

#### At Home (16:50 PM):
- **Whether to travel?** Yes.
- **Why travel?** I need to buy groceries and essentials for the
    household. It is a convenient time to head out now.
- **When to travel?** At 16:50 PM.
- **Where to travel?** A nearby grocery store.
- **How to travel?** Drive a van for personal convenience and to
    carry purchased goods.

#### At First Shopping Location (17:20 PM):
- **Whether to travel?** Yes.
- **Why travel?** After finishing my shopping at the first
    location, I need to visit another store for additional items
    that were unavailable at the first location.
- **When to travel?** Around 17:20 PM.
- **Where to travel?** Another nearby store.
- **How to travel?** Drive the van.

#### At Second Shopping Location (17:45 PM):
- **Whether to travel?** Yes.
- **Why travel?** After completing my shopping errands, I plan to
    participate in volunteer activities scheduled for the evening.
- **When to travel?** At 17:45 PM.
- **Where to travel?** The volunteer activity location.
- **How to travel?** Drive the van.

---

### Complete Travel Plan Table:

|Place Visited           |Arrival Time|Departure Time|Location Type|
|------------------------|------------|--------------|-------------|
|Home                    | 12:00 AM   | 4:50 PM      | 1           |
|First Shopping Location | 5:05 PM    | 5:20 PM      | 11          |
|Second Shopping Location| 5:35 PM    | 5:45 PM      | 11          |
|Volunteer Activity      | 5:50 PM    | 11:59 PM     | 5           |
```

## A.4 THE USE OF LARGE LANGUAGE MODELS

We use LLMs as a general-purpose auxiliary tool, primarily for the following purposes: a. Text polishing and grammar review. LLMs were used to improve the clarity, grammatical accuracy, and fluency of certain passages. b. Syntactic and lexical optimization: Without changing the core ideas and scientific content, LLM assisted in optimizing sentence structure and vocabulary selection.

