# OpenReview forum: "TravelReasoner: Reasoning-Augmented Travel Survey Simulations with Large Reasoning Models"
_ICLR.cc/2026/Conference — ICLR 2026 Conference Withdrawn Submission_

### Official Review · Reviewer_nmSy · 2025-10-28

**Soundness:** 2
**Presentation:** 2
**Contribution:** 2
**Rating:** 4
**Confidence:** 3

**Summary:**

This work builds reasoning data and reasoning models to enhance the accuracy of travel survey's accuracy. Travel survey is hard to obtain, so gaining more understandings on existing data is important, which is the motivation for introducing reasoning functions of LLMs to complete this task.

**Strengths:**

1. The motivation works for me: travel survey data are not only hard to obtain, but also can be very inaccurate. Introducing reasoning can help not only to understand more into existing data, but also to extract the reasonable parts of the existing data.
2. The designed method sounds for me: I believe the whether-why-where-when-how metrics can accurately catch reasonable travel, which helps to build reasonable travel models.
3. Quantitative results prove the effectiveness as well as the generalization abilities of the proposed method.

**Weaknesses:**

1. *Benefits of Reasoning are not demonstrated in detail.* While quantitative results suggest benefits from reasonings, qualitative examples are missing in the paper, which significantly undermine the claim. It is very hard to understand **why** the reasoning process is important, without seeing the reasoning itself, and without step-by-step study on **what information** is gained through reasoning and **how it is used** for enhancing quantitative metrics. I strongly suggest the authors adding a thorough step-by-step analysis to show the reasoning itself and its benefits.
2. Minor issue: number 1.93 in table 3 is not underlined.

**Questions:**

1. As the authors show in Table 1, with few shots, reasoning model seems not to out-perform the base model. In this case, why not using non-reasoning models with CoT shots? This seems not weakening the CoT data's benefits claimed.
2. I hope the authors can provide details on how the NHTS dataset is used for training and evaluations respectively, and whether information leak exists in any phase. Please note that my current evaluation is based on the case where no information leak exists. If this is incorrect, I will accordingly adjust my evaluations.

**Details Of Ethics Concerns:**

No ethics concerns found.

---

### Official Review · Reviewer_ZzpU · 2025-10-31

**Soundness:** 2
**Presentation:** 2
**Contribution:** 1
**Rating:** 2
**Confidence:** 4

**Summary:**

This paper proposes TravelReasoner, a reasoning-augmented framework for travel survey simulation based on Large Reasoning Models. The authors introduce a new dataset, Chain-of-Trips, and design a two-stage fine-tuning pipeline to enhance reasoning consistency and behavioral realism in simulated travel behaviors.

**Strengths:**

1. The design of the itinerary chain dataset—which connects reasoning traces with structured travel schedules—is conceptually sound and could prove valuable for future research.

2. The authors evaluate model performance across multiple cities and metrics (AverLoc, TimeCons, EditDis), reflecting a thorough consideration of empirical details.

3. The algorithm is clearly described, and the writing is accessible.

**Weaknesses:**

1. The primary weakness of this work lies in its lack of novelty. The core contribution essentially constitutes a straightforward application of existing reasoning-enhanced LLM training techniques (Chain-of-Thought plus Supervised Fine-Tuning) to a domain-specific dataset. The proposed two-stage pipeline does not introduce conceptual innovations. Moreover, the LRM is treated as a black box, with no novel architecture or training objective introduced.

2. The experimental comparisons are relatively weak, and the evaluation metrics appear inadequate—particularly for assessing the quality of reasoning paths, for which precise evaluation remains challenging.

**Questions:**

1. How does TravelReasoner differ algorithmically from a simple combination of Chain-of-Thought fine-tuning and reward-based filtering?

2. Why was reinforcement learning not adopted, especially given its prevalence in LRMs?

3. How does this task fundamentally differ from general travel planning?

4. Were comparisons conducted with planning-based or agentic reasoning models, such as ReAct?

---

### Official Review · Reviewer_a5zo · 2025-11-01

**Soundness:** 3
**Presentation:** 3
**Contribution:** 3
**Rating:** 4
**Confidence:** 2

**Summary:**

The paper introduces TravelReasoner, a framework that uses Large Reasoning Models (LRMs) to simulate human travel surveys. It also proposes a dataset which is constructed from real-world National Household Travel Survey (NHTS) data to capture the multi-step reasoning annotations behind travel decisions. The dataset is used with a two-stage post-training pipeline to improve the LRM's in-domain reasoning capabilities and the fidelity of the generated travel behaviors. Experimental results demonstrate significant improvements over baselines.

**Strengths:**

The paper is significant that an effective simulation tool could be highly valuable for urban planning and transportation modeling. The core idea of shifting simulation from just predicting sequences to modeling the underlying reasoning is a novel and valuable reframing of the problem. The paper is clearly written, and the proposed framework is well-organized.

**Weaknesses:**

1. The paper claims to model reasoning, but the quantitative evaluations (location and time consistency) only measure the outcome (the travel table). There is no empirical evaluation of the quality, plausibility, or faithfulness of the "Travel Reasoning Chain" itself.
2. The CHAIN-OF-TRIPS dataset lacks of human verifications on the quality, statistics of human checks should be provided.
3. For the 2-stage training pipeline, why don't you consider reinforcement learning finetuning on the LRM model to diversify the output (e.g., using GRPO).

**Questions:**

See weaknesses.

---

### Official Review · Reviewer_11RA · 2025-11-01

**Soundness:** 3
**Presentation:** 3
**Contribution:** 3
**Rating:** 6
**Confidence:** 2

**Summary:**

The paper proposes TravelReasoner, a reasoning-augmented framework for synthetic travel survey simulation. The core idea is to move from “fit an LLM to travel diaries” to “teach an LLM/LRM to reason about travel decisions, positing that realistic mobility emerges from stepwise choices of whether/why/when/where/how to travel. The authors create a Chain-of-Trips dataset from the 2017 NHTS by using GPT-4o to convert real trip records and user profiles into first-person reasoning traces and structured tables; implement a two-stage post-training pipeline, from B-SFT  to E-SFT, using LoRA on DeepSeek-R1-Distill-Llama-8B; and evaluate on multiple U.S. cities and cross-city transfer. The approach outperforms strong LLM-based mobility baselines in location consistency and temporal consistency, while generating interpretable reasoning traces for planners.

**Strengths:**

1. The paper makes a good conceptual move: instead of treating a travel diary as a flat sequence of locations, it recasts the diary as a chain-of-thought (CoT) over mobility decisions, whether/why/when/where/how. This is a nontrivial reframing because travel behavior research has long recognized that constraints and motivations are hierarchical, but LLM-based simulators usually ignore it.
2. The experimental section compares against multiple recent LLM mobility baselines, and TravelReasoner is consistently first or second on core metrics across multiple cities. This shows the method is not overfitting to one locale.
3. Synthetic mobility data is expensive to collect, and real survey data has low response rates. If a reasoning-augmented LLM can produce auditable, interpretable travel chains, that is practically valuable to transportation modeling, agent-based simulation, smart-city scenarios, and even data augmentation for scarce subpopulations.
4. The overall presentation is clear and easy-to-follow.

**Weaknesses:**

1. Data-generation reliance on GPT-4o is under-analyzed. This introduces a teacher–student bias: the evaluation may partly measure how well the student reproduces GPT-4o’s style, not how well it models human travel reasoning.
2. There is no human or expert-based assessment of the reasoning chains themselves (the core claimed contribution). We see that the model can output thoughts, but we do not know if these thoughts are actually consistent.

**Questions:**

1. What fraction of GPT-4o generations violated your own Instruction rules (non-monotonic times, missing “home,” wrong location code), and how were those handled? A short table here would make the dataset more credible.
2. Since GPT-4o is used to create the reasoning style, and the student LRM is trained to mimic it, how do you ensure you are not just distilling GPT-4o’s formatting? Have you tried training without the reasoning text but with the travel table only, to see how much of the gain is from explicit reasoning supervision?

---

### Note · Authors · 2025-11-22

I have read and agree with the venue's withdrawal policy on behalf of myself and my co-authors.